# Periodontitis Is Associated with Consumption of Processed and Ultra-Processed Foods: Findings from a Population-Based Study

**DOI:** 10.3390/nu14183735

**Published:** 2022-09-10

**Authors:** Luisa Schertel Cassiano, Marco A. Peres, Janaína V. S. Motta, Flávio F. Demarco, Bernardo L. Horta, Cecilia C. Ribeiro, Gustavo G. Nascimento

**Affiliations:** 1Section for Periodontology, Department of Dentistry and Oral Health, Aarhus University, 8000 Aarhus, Denmark; 2National Dental Research Institute Singapore, National Dental Centre Singapore, Singapore 168938, Singapore; 3Oral Health ACP, Health Services and Systems Research Programme, Duke-NUS Medical School, Singapore 168938, Singapore; 4Graduate Program in Epidemiology, Federal University of Pelotas, Pelotas 96020-220, Brazil; 5Graduate Program in Dentistry, School of Dentistry, Federal University of Pelotas, Pelotas 96015-560, Brazil; 6Department of Dentistry II, Federal University of Maranhão, São Luis 65085-580, Brazil

**Keywords:** periodontitis, food consumption, diet

## Abstract

The association between periodontitis and lifestyle factors has been widely investigated. However, an association between periodontitis and dietary patterns has not been explored. Therefore, this study investigated the association between periodontitis and food consumption among a Southern Brazil population. Data from the 1982 Pelotas Birth Cohort were used (*n* = 537). The exposure, periodontitis, was clinically measured and classified using the AAP/CDC system, then two latent variables were defined: ‘initial’ and ‘moderate/severe’ periodontitis. The consumption of in natura, processed, and ultra-processed foods (NOVA classification) was the outcome and measured in calories using the food frequency questionnaire (FFQ). Confounders were sex, maternal education, smoking status, xerostomia, and halitosis. Data were analyzed by structural equation modeling. ‘Initial’ periodontitis was associated with a higher consumption of in natura food (standardized coefficient (SC) 0.102; *p*-value = 0.040), versus processed (SC 0.078; *p*-value = 0.129) and ultra-processed (SC 0.043; *p*-value = 0.400) foods. ‘Moderate/severe’ periodontitis was associated with higher consumption of ultra-processed foods (SC 0.108; *p*-value = 0.024), versus processed (SC 0.093; *p*-value = 0.053) and in natura (SC 0.014; *p*-value = 0.762) foods. ‘Moderate/severe’ periodontitis appears to be associated with the consumption of processed and ultra-processed foods.

## 1. Introduction

Food consumption has been studied in different fields: nutrition, psychology, social science, and marketing. In a didactic manner, factors affecting food choices can be grouped into three main groups that comprise (1) intrinsic (e.g., color, texture) and extrinsic (e.g., packaging) food-related features; (2) individual differences in biological (e.g., hunger, ability to smell and taste), physical (e.g., skills of cooking, access), psychological (e.g., stress), cognitive (e.g., experiences), and social (e.g., family, friends) levels; and (3) society-related features, including cultural, economic, and political aspects [1,2].

Even though food consumption has a behavioral and environmental background, sensory aspects related to the food are crucial in food choice [3]. A review concluded that visual and odor cues guide food consumption and memory for eating, while tastes and textures affect meal size and satisfaction. Another review indicated that the food smell and taste play a relevant role in a macronutrient sensing system and food consumption, as the former induces appetite. At the same time, the latter contributes to satisfaction based on the eating rate and duration of food exposure in the mouth [4]. Ultra-processed foods tend to be highly appealing as, due to their high degree of processing and addition of sugar and saturated fat, they become very palatable [5,6]. Thus, based on previous pleasant experiences and sensory aspects of ultra-processed foods, it is not surprising that ultra-processed foods are frequently perceived as tastier than in natura food and, therefore, often preferred [7,8].

Periodontitis is an inflammatory disease that destroys the supporting tissues of the teeth. While the local biofilm may influence the onset and progression of the periodontitis, host factors such as lifestyle and genetics appear to play a significant role [9]. The disease’s common signs and symptoms include gingival bleeding and pocket formation, and halitosis and tongue coating can also be associated with periodontal disease [10]. While the association between periodontal disease and lifestyle factors, such as smoking and alcohol consumption, has been widely investigated, the possibility that periodontitis can influence dietary patterns must be further explored. At an advanced stage, periodontitis can lead to tooth loss and partial or complete edentulism, with the need for implant treatment [11,12].

Previous studies have demonstrated an association between added sugar [13], carbohydrate consumption [14], and periodontitis. In a recent population-based study, Costa et al., 2022 reported that an unhealthy diet with higher consumption of fast food and manufactured products was associated with initial and moderate periodontitis in Brazilian adolescents [15]. Evidence shows that a diet rich in salads, fruits, and vegetables appears to be associated with less attachment loss in American adults [16]. In contrast, a Western diet based on ultra-processed food appears to increase the risk of self-reported periodontitis [17]. Meanwhile, a study using data from the NHANES failed to encounter an association between ultra-processed food intake and periodontitis among American adults [18]. However, the NHANES survey uses 24-h food recalls, which may not truly reflect one’s dietary pattern and may lead to social desirability bias. Moreover, their periodontal classification relied on an observed variable, which might have influenced their findings. Similar results were seen in previous studies investigating periodontitis and other multidimensional conditions [19]. Thus, there are several ways to enhance the knowledge about the relationship between periodontitis and food consumption.

This study, therefore, aimed to investigate the possible influence that periodontitis can have on food consumption among adults in the Southern Brazilian city of Pelotas.

## 2. Materials and Methods

### 2.1. Pelotas Birth Cohort

In 1982, the three maternity hospitals of Pelotas in southern Brazil were visited daily, and all births were identified. Those liveborn infants whose parents lived in the city’s urban area were examined (*n* = 5914), and the mothers were interviewed. Interviews covered aspects of socioeconomic conditions, dietary habits, smoking, alcohol consumption, and health conditions. These individuals have been followed on several occasions (Figure 1). More details were previously published [20].

In 1997, at the age of 15, a systematic random sample of 70 (27%) of the 259 census tracts located within the city limits was selected; all the households included in these tracts were visited, and 900 adolescents were selected for the oral health study (OHS)-97. Of the 900 participants, 888 individuals (98.7%) were orally examined. In 2006, at the age of 24, participants of the OHS-97 were invited for a new oral health examination, and a total of 720 (81.1%) were followed up. In 2013 (OHS-13), at the age of 31, all study participants of the OHS-97 were again contacted for a third oral health assessment. Data used in this study originate from the OHS-13 [21], comprising 537 participants. The flowchart of the 1982 Pelotas birth cohort study is displayed in Figure 1.

This observational study used the Strengthening the Reporting of Observational Studies in Epidemiology (STROBE) guidelines to guide its reporting. The Ethics Committee of the Federal University of Pelotas approved all assessments (IRB#19551713.9.0000.5317), and informed consent was obtained from all participants. The study was conducted following the Declaration of Helsinki.

### 2.2. Food Consumption: Outcome

Food consumption was measured through the food frequency questionnaire (FFQ), which contained self-reported information on the frequency, amount, and portion size of in natura, processed, and ultra-processed food consumed in the past 12 months. Based on these data, the number of calories consumed was calculated. Data were collected at the age of 30 years. The FFQ was adapted from Sichieri [22] and contained 85 food items divided into two components: quantitative, which consisted of evaluating food portions and preparation, and qualitative, which assessed the frequency of food consumption [23]. The FFQ is a cost-effective way of collecting long-term dietary intake data from several respondents [24]. For this reason, several cohort studies, such as the Framingham cohort and the EPIC cohort, have used the FFQ as a tool for diet measurement [25,26].

The NOVA classification system [27] was used to classify food according to the degree of processing, as follows: unprocessed or minimally processed (in natura) foods (group 1); processed foods used as ingredients in cooking preparations by the food industry (group 2); processed foods (group 3); and ultra-processed foods or food products (group 4). Groups 2 and 3 were grouped into ‘processed foods’ in this study.

### 2.3. Periodontitis: Exposure

All teeth were examined for periodontal disease (bleeding on probing—BOP, probing depth—PD, and clinical attachment loss—CAL) and other oral conditions. Six previously calibrated dentists who underwent theoretical and practical training on 25 individuals performed the oral examination. The lowest intraclass correlation coefficient for pocket depth and CAL was 0.85. Clinical oral examination was carried out at the participants’ homes following the biosafety procedures recommended by the World Health Organization for epidemiological surveys. A headlight, dental mirror, and PCP2 periodontal probe with 2-mm banding were used (Hu-Friedy PCP-2, Rotterdam, the Netherlands). Periodontal examinations consisted of full-mouth probing at six sites per tooth in all present teeth, excluding third molars.

Periodontitis was classified using the American Academy of Periodontology and the Center for Diseases Control and Prevention (AAP/CDC) classification and later dichotomized into two different variables: health (reference category) vs. any periodontitis; and health/mild periodontitis (reference category) vs. moderate/severe periodontitis [28]. Another publication used this approach using the 1982 Pelotas birth cohort data [14].

Additionally, two latent variables were obtained to better reflect the multidimensionality of periodontitis. These variables were created based on an exploratory factor analysis, a statistical method where observed variables are grouped into latent variables that share a common variance and were not observed [29]. The two latent variables are defined as follows:-‘Initial’ Periodontitis comprises sites with a clinical attachment level of <4 mm, probing depth < 4 mm, and bleeding on probing.-‘Moderate/Severe’ Periodontitis comprises sites with clinical attachment loss > 4 mm and <7 mm, and probing depth > 4 mm and <7 mm.

Factor loadings for each latent periodontal variable are available in Appendix A. Variables loading on both factors were included in neither factor as they could not discriminate a specific factor variable.

### 2.4. Covariates

Sex was measured at birth (1982) and treated as a dichotomous variable. Maternal education level was measured (through the question “how many years of study have you completed?”) at birth and later dichotomized into completed high school or above and not completed high school or below. The following variables were measured at the age of 30 and dichotomized as follows: smoking status (never smoked, and former/current smokers), self-reported xerostomia (“Have you had a feeling of dry mouth in the past six months?”/yes or no), and self-reported halitosis (“Do you think you have bad breath?”/yes or no).

Evidence suggests the association between xerostomia and periodontitis [30] and xerostomia and food consumption [31]. Additionally, halitosis appears to be positively associated with periodontitis [10] and food intake [32]. Therefore, these variables were included in the analysis.

### 2.5. Analytical Approach

Descriptive analysis for all variables in the study was performed, and the percentages for categorical variables and median and interquartile range for continuous variables were obtained.

The analysis was performed with structural equation modeling (SEM). SEM is a multivariable analysis tool to model complex relations among variables, allowing effect decomposition and explicitly identifying direct and mediated relationships [33]. The initial exploratory analysis was done to determine the latent variables for periodontitis (Appendix A). Then, the confirmatory factor analysis was carried out to check for the model fitting, showing that the two proposed constructs for periodontitis had a satisfactory fit. Next, the proposed structural model was tested. Modification indices (MIs) suggested by the statistical software were estimated and implemented when necessary and supported by previous literature. The removal of non-significant pathways was also performed to ensure the model parsimony. These steps aimed to improve the structural model by exploring and generating new models. Furthermore, an interaction analysis was carried out to test the effect of the interaction between ‘initial’ and ‘moderate/severe periodontitis’ and halitosis on food consumption by including a variable containing the cross-product between halitosis and periodontitis in the final structural model. A model fit was assessed using the root mean square error of approximation (RMSEA < 0.08), the comparative fit index (CFI > 0.95), and the Tucker-Lewis index (TLI > 0.95) [34]. Descriptive analysis was performed using Stata 16.1 (StataCorp., College Station, TX, USA), and SEM was performed using MPlus 8.0 (Muthén and Muthén, Los Angeles, CA, USA).

The standardized analysis was used to get standardized coefficients based on the standard deviation. Thus, a standardized coefficient between 0.1 and 0.2 indicates a weak association; between 0.3 and 0.4, a moderate association; and ≥0.5, a strong association [4].

Additionally, we have conducted a sensitivity analysis with the observed variables for periodontitis based on the AAP/CDC classification and the exposure to investigate if our findings follow the same pattern as previously published by Nascimento et al., 2019 using SEM [17].

## 3. Results

All individuals that participated in the OHS-97 (at 15 years) (*n* = 888) were invited to participate in the OHS-13 (at 31 years). Despite the 40% attrition rate, the population of this study remains representative of the original cohort (Table 1). In this study (*n* = 537), the population consisted of 50.6% males, and any measured periodontitis was present in 38% of the population following the AAP/CDC classification. Halitosis was present in 37% of the population, while xerostomia was present in 22%. A total of 18% of the participants were smokers or former smokers. Regarding food consumption, a median of 1029.98 kcal/day was from in natura foods, 181.90 kcal/day from processed foods, and 413.13 kcal/day from ultra-processed foods. A detailed description of the population included in this study is presented in Table 1.

The results show that ‘initial’ periodontitis was associated with a higher calorie consumption of in natura food (standardized coefficient (SC) 0.102; *p*-value = 0.040), when compared to processed (SC 0.078; *p*-value = 0.129) and ultra-processed (SC 0.043; *p*-value = 0.400) foods (Table 2). On the other hand, ‘moderate/severe’ periodontitis was associated with a higher calorie consumption of ultra-processed foods (SC 0.108; *p*-value = 0.024), when compared to in natura (SC 0.014; *p*-value = 0.762) and processed (SC 0.093; *p*-value = 0.053) foods (Table 3). An indirect effect between periodontitis and food consumption mediated by halitosis was not detected. Finally, interaction analysis did not reveal any effects from the interaction between periodontitis and halitosis (Appendix B).

## 4. Discussion

Our findings suggest that the severity of periodontitis is associated with the processing level of consumed food. ‘Moderate and severe’ periodontitis had a stronger association with the consumption of processed and ultra-processed foods, while ‘initial’ periodontitis was associated with higher consumption of in natura foods in a southern Brazilian population. Based on these findings, we speculate whether periodontitis may impair the senses of taste, leading to higher consumption of highly palatable foods. Halitosis did not interact nor mediate the effect between periodontitis and food consumption.

One of the limitations this study faces is the 40% attrition rate among the participants of the first oral health assessment (OHS-97) observed in this cohort 31 years after its establishment. This attrition rate is similar to other studies found in the literature [35,36,37]. Despite the observed attrition rate, the OHS-13 (31 years old) remains representative of the original cohort, as previously demonstrated [14,38,39,40]. Additionally, one may speculate to which extent our sample size has influenced our results. While the sample size could have underestimated our findings, this would have been related to ‘moderate/severe’ periodontitis and not to ‘initial’ periodontitis, as the latter is our largest group. However, it is possible to presume that the sample size could have affected the precision of our estimates, not the direction of the association. Therefore, in the case of ‘moderate/severe’ periodontitis, it could be assumed that there would have been more minor standard errors had the sample size been larger. Moreover, even though we have considered relevant confounders in our analysis, it is impossible to rule out residual confounding inherent to observational studies. Despite the availability of methods to assess unmeasured confounding when structural equation modeling is used, it was not possible to apply it in our study due to our limited sample size [41]. Finally, although one may question the validity of our findings, since this study originates from a middle-sized city in Southern Brazil, the 1982 Pelotas Birth Cohort is the longest birth cohort study in a low-middle-income country with oral health data. Numerous cutting-edge studies and public health policies derive from this birth cohort data, reinforcing the relevance of our study [14,39,42,43].

This study followed the Nova Food Classification System [27]. This classification groups food according to its nature, extent, and purpose: in natura (or unprocessed) or minimally processed foods; processed culinary ingredients; processed foods; and ultra-processed foods. However, the FFQ used in our study was not specifically tailored to the NOVA food classification system. Hence, non-differential classification error cannot be ruled out, potentially leading to the underestimation of the magnitude of the associations observed in our study. Moreover, the FFQ applied in our study was designed to evaluate food consumption in the cohort population and not to assess the degree of food processing. Thus, as it was impossible to obtain information on food preparation, food items from the FFQ were categorized more conservatively. For instance, a culinary preparation such as lasagna was considered unprocessed food, whereas bread (whole or white) was considered processed [44]. This is further corroborated by Petrus et al. (2021), who state that the NOVA system may complicate and overlook the simple and basic health factors regarding food decisions and choices [45]. The authors also note that the system is not accurately defining foods. Nevertheless, this is the most used system currently used, and for comparison purposes, it is a valuable instrument.

Among our study’s significant strengths, its methodological approach, including the use of latent variables for periodontitis, should be stressed. Factor analyses revealed that periodontitis comprised two dimensions, ‘initial’ and ‘moderate/severe’ periodontitis. ‘Initial’ periodontitis consists of bleeding on probing, shallow pockets, and mild attachment loss, versus ‘moderate/severe’ periodontitis, represented by deep periodontal pockets and moderate-to-severe attachment loss. While the consumption of ultra-processed foods was associated with ‘moderate/severe’ periodontitis, the consumption of in natura foods was associated with ‘initial’ periodontitis, corroborating our hypothesis. On the other hand, no association was identified when periodontitis was treated as an observed variable (AAP/CDC classification) in the structural equation models, similar to a previous study published on this population [19]. To our knowledge, this is the first study investigating the impact of periodontitis on food consumption using latent variables in structural equation models. Finally, the quality of the data used in this study is worth mentioning. The 1982 Pelotas Birth Cohort is the largest and longest cohort with oral-health outcomes in low-middle-income countries. The use of clinical data for periodontitis, collected with high inter-examiner reliability, is a strength of this study.

We hypothesize that periodontitis influences food consumption by impairing chemosensory perception through oral inflammation. A biologically plausible model relies on three parameters: inflammatory molecules and increased proteolytic activity from oral inflammatory lesions could induce apoptosis of taste buds on the tongue, which leads to tasting distortion; tongue-coating biofilm could act as a physical barrier preventing contact between taste molecules and the tongue receptors; and, halitosis could distort smell sensitivity, as bacteria putrefy organic compounds from periodontal lesions and the tongue. In a previous systematic review by Schertel Cassiano et al., 2020, the authors found that halitosis seems to be associated with impaired senses of smell and taste [46]. Therefore, it is possible to assume that oral inflammation can impair chemosensory perception [44] and, as a result, influence food consumption [3,4,47].

In addition, taste and smell perception can also influence food choice and consumption. While taste is mainly considered a nutrient-sensing system, smell has functions other than eating, including activating approach-avoidance behavior to potential environmental hazards for foods [4]. In a study with 199 Finnish adults, Puputti et al., 2019, reported that all modality-specific taste sensitivities were related to some consumption behavior, demonstrating that food consumption habits were related to taste sensitivity [3]. On a similar note, Jayasinghe et al., 2017, found an association between sweet taste perception and sugar intake, demonstrating that taste perception is likely associated with food consumption [47]. Hence, it is possible to speculate that factors affecting the senses of smell and taste, such as possible periodontitis, may indirectly affect food choice and consumption.

While our hypothesis is biologically plausible, one cannot rule out the possibility that these conditions, periodontitis, and the consumption of ultra-processed foods coexist without a clear causal relationship between them. In addition, as we used cross-sectional data nested in a cohort study, it is not possible to eliminate the chance that high consumption of ultra-processed food might have led to periodontitis as the latter has been associated with added sugar and carbohydrates [48]. However, as only cases of ‘moderate/severe’ periodontitis were related to consumption of processed/ultra-processed foods, it is possible to speculate that a long exposure period to oral inflammation would be required until a food preference change is noted. Future studies ought to further elucidate this relationship. Regardless, our results shed light on an association between periodontitis and food consumption and indicate the need for a more comprehensive approach when treating individuals with periodontitis. Furthermore, if our assumption holds, the treatment of periodontitis could improve one’s ability to taste food, impacting one’s quality of life and other systemic conditions related to the high consumption of ultra-processed food.

## 5. Conclusions

In conclusion, we believe that the association of periodontitis with food consumption possibly mediated by taste disorders is a broad field to be explored. The evidence supports the need to understand periodontitis as a complex condition, and to reinforce the need for a multidisciplinary treatment approach.

## Figures and Tables

**Figure 1 nutrients-14-03735-f001:**
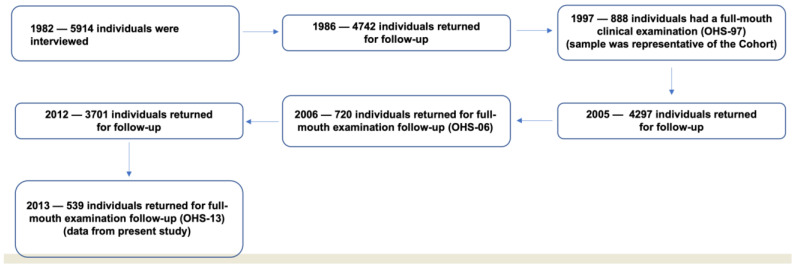
The flowchart of the 1982 Pelotas Birth Cohort Study.

**Table 1 nutrients-14-03735-t001:** Descriptive frequency of the variables used in this study (*n* = 537) and comparison between the original cohort population and the participants of OHS-13 (2013).

Variables	Distribution N (%) OHS-13 (2013) (*n* = 537)	Distribution N (%) Original Cohort (1982) (*n* = 5914)
Sex (Male)	272 (50.7)	53%
Maternal education (high school or above)	223 (41.5)	43.9%
Periodontitis (initial, moderate, and severe)	205 (38.2)	-
Halitosis	200 (37.2)	-
Xerostomia	117 (21.8)	-
Smoking	95 (17.7)	-
Food Consumption (calories)	Median (Interquartile Range)	Median (Interquartile Range)
In natura	1029.98 (707.6; 1513.6)	1124.18 (797.5; 1609.9)
Processed	181.90 (84.5; 331.6)	206.6 (112.7; 345.2)
Ultra-processed	413.13 (230.0; 666.0)	460.5 (284.0; 725.1)

**Table 2 nutrients-14-03735-t002:** Standardized effects of food consumption and initial periodontitis among adults enrolled in the 1982 Pelotas birth cohort.

Pathways	Standardized Coefficient (SC)	Standard Error (S.E.)	Estimate/S.E.	*p*-Value
In natura on ‘initial’ periodontitis	0.102	0.050	2.051	0.040
Processed on ‘initial’ periodontitis	0.078	0.052	1.519	0.129
Ultra-processed on ‘initial’ periodontitis	0.043	0.051	0.842	0.400

**Table 3 nutrients-14-03735-t003:** Standardized effects between food consumption and moderate/severe periodontitis among adults enrolled in the 1982 Pelotas birth cohort.

Pathways	Standardized Coefficient (SC)	Standard Error (S.E.)	Estimate/S.E.	*p*-Value
In natura on ‘moderate/severe’ periodontitis	0.014	0.048	0.303	0.762
Processed on ‘moderate/severe’ periodontitis	0.093	0.048	1.937	0.053
Ultra-processed on ‘moderate/severe’ periodontitis	0.108	0.048	2.261	0.024

## Data Availability

The data that support the findings of this study are available on request from the corresponding author. The data are not publicly available due to privacy or ethical restrictions.

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
