# Peer review of "Periodontitis Is Associated with Consumption of Processed and Ultra-Processed Foods: Findings from a Population-Based Study"

_nutrients, 2022, doi:10.3390/nu14183735_

Round 1

Reviewer 1 Report

Introduction and discussion should be improved with the suggested references

Materials and methods are well described and pertinent

Results are clearly described and very coherent with the materials and methods

Conclusions are pertinent and fit with the presented result

However, the suggested references are for the following topics:

And Authors should describe how innovative rihabilitatione techniques have to be considered in this PERIODONTAL patientS, to evaluate, because the possibility of having to perform the extraction and rehabilitation of the prosthetic implant of the non-recoverable dental element should also be considered

(PubMed ID: PMID: 34425665 DOI: 10.23812/21-4supp1-9)

(PMID: 33810379-PMCID: PMC8037328-DOI: 10.3390/ijerph18073449)

Author Response

We thank the reviewer for this comment. The suggested references have been added to the third paragraph on the introduction (lines 56 and 57).

Reviewer 2 Report

General comments

As an association between periodontitis and dietary patterns has not been explored, this study investigated the relationship between periodontitis and food consumption within the Southern Brazil population using the data from the 1982 Pelotas Birth Cohort (n = 537). This study demonstrates that 'Moderate/severe' periodontitis appears to be associated with the consumption of processed and ultra-processed foods.

Specific comments

The aim of this study appeared to be important in understanding as association between periodontitis and the pattern of food consumption. However, the contents along with the research focus in this paper tended to close to a ‘report’ rather than an ‘article’. Overall, the results shown in this study don’t reflect a broad and deep understanding on the relationship between the levels of periodontic disorders and the food consumption. This is because that the results were derived only from a specific region, namely the Southern Brazilian city of Pelotas. There was also no any comparative results with other regions. Together, the reviewer considers that this paper is to be a minor report, and also have no a scientific significance at the current form. 

Author Response

We thank the reviewer for this comment. The 1982 Pelotas Birth Cohort is the longest ongoing birth cohort with oral health data in a low-middle income country. Such a birth cohort is of relevance not only for oral health but numerous cutting-edge studies and public health policies derive from the data collected there. A quick search on PubMed reveals more than 200 articles published using this cohort data. Please see a few examples below:

Barros FC, Victora CG, Barros AJ, Santos IS, Albernaz E, Matijasevich A, Domingues MR, Sclowitz IK, Hallal PC, Silveira MF, Vaughan JP. The challenge of reducing neonatal mortality in middle-income countries: findings from three Brazilian birth cohorts in 1982, 1993, and 2004. Lancet. 2005 Mar 5-11;365(9462):847-54. doi: 10.1016/S0140-6736(05)71042-4. PMID: 15752528.

Victora CG, Horta BL, Loret de Mola C, Quevedo L, Pinheiro RT, Gigante DP, Gonçalves H, Barros FC. Association between breastfeeding and intelligence, educational attainment, and income at 30 years of age: a prospective birth cohort study from Brazil. Lancet Glob Health. 2015 Apr;3(4):e199-205. doi: 10.1016/S2214-109X(15)70002-1. PMID: 25794674; PMCID: PMC4365917.

Horta BL, Barros FC, Lima NP, Assunção MCF, Santos IS, Domingues MR, Victora CG; Pelotas Cohorts Study Group. Maternal anthropometry: trends and inequalities in four population-based birth cohorts in Pelotas, Brazil, 1982-2015. Int J Epidemiol. 2019 Apr 1;48(Suppl 1):i26-i36. doi: 10.1093/ije/dyy278. PMID: 30883661; PMCID: PMC6422063.

Wehrmeister FC, Victora CG, Horta BL, Menezes AMB, Santos IS, Bertoldi AD, da Silva BGC, Barros FC; Pelotas Cohorts Study Group. Hospital admissions in the first year of life: inequalities over three decades in a southern Brazilian city. Int J Epidemiol. 2019 Apr 1;48(Suppl 1):i63-i71. doi: 10.1093/ije/dyy228. PMID: 30883660; PMCID: PMC6422058.

In addition, data from food consumption and other systemic health outcomes have been recently published in high-impact journals, as follows:

Silva Dos Santos F, Costa Mintem G, de Oliveira IO, Horta BL, Ramos E, Lopes C, Gigante DP. Consumption of ultra-processed foods and interleukin-6 in two cohorts from high- and middle-income countries. Br J Nutr. 2022 Feb 21:1-28. doi: 10.1017/S0007114522000551. Epub ahead of print. PMID: 35184789.

Werneck AO, Costa CS, Horta B, Wehrmeister FC, Gonçalves H, Menezes AMB, Barros F, Monteiro CA. Prospective association between ultra-processed food consumption and incidence of elevated symptoms of common mental disorders. J Affect Disord. 2022 Sep 1;312:78-85. doi: 10.1016/j.jad.2022.06.007. Epub 2022 Jun 9. PMID: 35691417.

In oral health, several studies using the 1982 Pelotas Birth Cohort have also been published:

Demarco FF, Cademartori MG, Hartwig AD, Lund RG, Azevedo MS, Horta BL, Corrêa MB, Huysmans MDNJM. Non-carious cervical lesions (NCCLs) and associated factors: A multilevel analysis in a cohort study in southern Brazil. J Clin Periodontol. 2022 Jan;49(1):48-58. doi: 10.1111/jcpe.13549. Epub 2021 Oct 26. PMID: 34545588.

Boscato N, Nascimento GG, Leite FRM, Horta BL, Svensson P, Demarco FF. Role of occlusal factors on probable bruxism and orofacial pain: Data from the 1982 Pelotas birth cohort study. J Dent. 2021 Oct;113:103788. doi: 10.1016/j.jdent.2021.103788. Epub 2021 Aug 21. PMID: 34425171.

Nascimento GG, Goettems ML, Schertel Cassiano L, Horta BL, Demarco FF. Clinical and self-reported oral conditions and quality of life in the 1982 Pelotas birth cohort. J Clin Periodontol. 2021 Sep;48(9):1200-1207. doi: 10.1111/jcpe.13512. Epub 2021 Jul 7. PMID: 34169558.

Schuch HS, Nascimento GG, Peres KG, Mittinty MN, Demarco FF, Correa MB, Gigante DP, Horta BL, Peres MA, Do LG. The Controlled Direct Effect of Early-Life Socioeconomic Position on Periodontitis in a Birth Cohort. Am J Epidemiol. 2019 Jun 1;188(6):1101-1108. doi: 10.1093/aje/kwz054. PMID: 30834447.

Nascimento GG, Peres MA, Mittinty MN, Peres KG, Do LG, Horta BL, Gigante DP, Corrêa MB, Demarco FF. Diet-Induced Overweight and Obesity and Periodontitis Risk: An Application of the Parametric G-Formula in the 1982 Pelotas Birth Cohort. Am J Epidemiol. 2017 Mar 15;185(6):442-451. doi: 10.1093/aje/kww187. PMID: 28174825.

Thus, one must assume that the data used in our study is far from being a minor report and brings a relevant contribution to the field. 

Additionally, we have added the information about the population (lines 215 and 216) and the following sentence (lines 234-38) to the discussion:

"Finally, although one may question the validity of our findings, since this study originates from a middle-sized city in Southern Brazil, the 1982 Pelotas Birth Cohort is the longest birth cohort study in a low-middle-income country with oral health data. Numerous cutting-edge studies and public health policies derive from this birth cohort data, reinforcing the relevance of our study42,43,44,45."

Reviewer 3 Report

Dear authors this is a very interesting study. 

Author Response

We thank the reviewer for this comment.

Round 2

Reviewer 2 Report

There are no more comments to authors.